# Mobile P2P-Based Skyline Query Processing over Delay-Tolerant Networks

**Kyoungsoo Bok [1], Sunyong Park [2], Jongtae Lim [2] and Jaesoo Yoo [2,\*]**

[1] Department of SW Convergence Technology, Wonkwang University, Iksandae 460, Iksan, Jeonbuk 54538, Korea; ksbok@wku.ac.kr
[2] Department of Information and Communication Engineering, Chungbuk National University, Chungdae-ro 1, Seowon-Gu, Cheongju, Chungbuk 28644, Korea; sy920112@naver.com (S.P.); efzotz@gmail.com (J.L.)
[\*] Correspondence: yjs@chungbuk.ac.kr; Tel.: +82-43-261-3230

**Abstract:** Skyline query-processing techniques considering various properties in peer to peer (P2P)-based services have become a recent topic of research. In this paper, we propose a new skyline query-processing scheme to improve the query-processing performance and accuracy in a mobile P2P service over delay-tolerant networks. The proposed scheme collects data on the query object from neighboring nodes and establishes a local skyline through static properties to reduce query-processing costs. To improve the query accuracy in a non-uniform distribution environment, the query-dissemination range is expanded by enforcing a query-dissemination range expansion. The performance evaluation conducted to verify the superiority of the proposed scheme demonstrates that it has a better performance compared to the existing schemes.

**Keywords:** skyline query; local skyline; query dissemination; filtering object; mobile P2P

## 1. Introduction

The existing Transmission Control Protocol/Internet Protocol (TCP/IP) based on end-to-end connectivity is difficult to apply in a long-latency, low data rate, and intermittent connectivity environment. A delay-tolerant network (DTN) is proposed to support network connection disruption even in environments where end-to-end connectivity is not guaranteed [1,2]. DTN is a new addition of bundles between the transmission layer and the application layer of the Open Systems Interconnection (OSI) 7 hierarchy structure, and performs communication in an environment that exists in non-connectivity through multi-hop communication. DTN uses a store-and-forward message transmission scheme to support node-to-node communication even in environments where end-to-end connectivity is not guaranteed [3,4]. DTN technology is being used in environments with high network disconnection such as sensor networks, mobile ad hoc networks, and vehicular ad hoc networks.

DTN is applied to various peer to peer (P2P) services because it can transmit data between nodes through multi-hop communication [5,6]. There are currently many ongoing studies on the topic of mobile P2P (MP2P) services that include mobility in the DTN, following the increased use of wireless devices such as smartphones, tablet PCs, and laptops resulting from advancements in mobile communication [7,8]. In contrast to existing P2P services where the bandwidth is stable and storage expansion is easy to manage, mobile P2P services use short-range wireless communication between mobile devices to configure an autonomous network [9–11]. Therefore, there are limitations in bandwidth, storage, computing power, battery, etc. Moreover, the data transmission and receipt route change frequently owing to constant changes in the network topology due to the mobility of nodes [12–16].

In order to provide the data requested by each node from P2P, data are collected and queries are processed through collaboration with neighboring nodes [17–20]. Query-processing schemes are being studied in order to process the search range considering the location, k-nearest neighbor search, reverse k-nearest neighbor search, etc. as nodes are continuously moving in DTN [21–24]. New studies are being conducted on top-k query-processing and skyline query-processing schemes, which can perform searches by considering two or more properties [25–30]. Skyline queries is used to provide objects with at least one or more superior properties in decision making or recommendation services that require multiple properties to be considered simultaneously. Skyline query-processing search objects with multiple properties that are not dominated by the properties of other objects [28,29]. Here, "objects that are not dominated" refer to objects with at least one property that is superior to the properties of another object. In other words, skyline query-processing removes only objects whose properties are not superior to other objects. For example, assume that a user is searching for nearby food that is affordable. If object A is 30 km away and costs 1000 won and object B is 20 km away but costs 2000 won, although object A is farther than object B, because the price is cheaper, it is not dominated by object B. Therefore, the search results will show both object B and object A. However, if all the properties of an object are dominated by other objects, it will not be shown in the search results.

In a DTN environment, each node with mobility acts as peer and performs skyline query using objects collected from neighboring nodes through short-range wireless communication. Since nodes continue to receive new objects, each node requires a high cost of communication and calculation for skyline query processing. Several studies have recently been conducted on effectively processing skyline queries in the mobile P2P services. Because such processing typically transmits object data from all nodes within a specific range to the node that made the query, unnecessary query processing costs are incurred. In order to resolve this issue, various filtering techniques, which can reduce skyline query-processing costs and improve performance, are being studied [30,31]. The volume of dominating region (VDR), which involves selecting the object that dominates the largest range in the skyline as the filtering object in order to filter many objects in a uniform distribution environment, is proposed in [30]. On the other hand, [31] proposed the distance between the points and line (DPL), in which objects are filtered by using the mode value to reduce communication costs in a non-uniform distribution environment. However, these existing schemes process local skyline requests by collecting object data for each node's query and merge the query results of each node in order to generate the final query-processing results. Because each node compares and filters all the query objects collected by the nodes, high calculation costs are incurred. Moreover, if the node distribution is non-uniform, the query dissemination cannot be processed appropriately and query accuracy will suffer.

In this paper, we propose a P2P-based efficient skyline query-processing scheme to improve the query-processing performance and accuracy over DTN. The proposed scheme generates candidate-filtering object groups by establishing a local skyline through data dissemination with neighboring nodes in order to reduce communication costs and improve query-processing speeds. It processes queries by expanding the query dissemination range to improve the query-processing accuracy in a non-uniform distribution environment. Because query dissemination is not required, the issue of reduced query-processing accuracy is eliminated.

This paper is organized as follows. Section 2 introduces the existing skyline query-processing schemes and Section 3 describes the proposed skyline query-processing scheme for mobile P2P services in DTN. Section 4 describes the performance evaluation results to show the superiority of the proposed scheme. Finally, Section 5 presents the conclusion and the direction for future work.

## 2. Related Work

In [19], a database management system called MOBIle DIscovery of Knowledge (MOBI-DIK) was proposed in Mobile P2P. In MOBI-DIK, each mobile node has a local database that stores and manages a collection of data items or reports. In MOBI-DIK, a ranking scheme based on supply and

demand is used. To process a query in mobile P2P, each node transmits directly reports and queries to neighbors, and they are propagated by multi-hop communications.

In [27], a top-k query-processing scheme was proposed to reduce the traffic and ensure the accuracy of query processing in mobile ad hoc networks. Each node has a routing table which consists of the ranking of scores of objects and the identifiers of nodes called the query address that queries are forwarded to acquire the necessary objects. Each node updates its own routing table when the link disconnection occurs and scores of objects are updated. The query node disseminates a query information attached ranks of objects that are required for the query address. If nodes receiving the query message store the requested objects, they reply the objects. When the score of an object is updated, the node that updated the score disseminates a message the other nodes which stores objects whose ranks have changed.

In [30], a disseminated skyline query-processing scheme was proposed to reduce the cost of communication and the execution time. To reduce the amount of data to be transferred, the query node disseminates the query information with spatial constraints and each node receiving the query information sends the query node a local skyline. The query node also calculates a local skyline after sending out the query information. To reduce local results that do not belong to the final skyline, VDR is used. The tuple called $tp_{flt}$ from the initial local skyline line on the query node is selected to filter out non-qualifying data. $tp_{flt}$ is updated during the query processing to increase the pruning potential.

In [31], an effective filtering scheme considering the data dissemination for skyline queries was proposed to reduce the communication cost. To reduce the communication cost under skewed distribution, a mode value is used. The frequencies of values on each dimension are measured to fine a mode value. The filtering data based on the mode values is selected. A mode point (MP) is defined to present the mode values of each dimension. The query node selects a local skyline object as the filtering object that has the smallest value of DPL, where DPL is the distance from a skyline object to line that passes the point o and MP. The query node disseminates the query information with filtering object and frequencies of values. Later, the query node merges the skyline results of other nodes and produces the final skyline results.

The existing skyline query-processing scheme generates a filtering object to perform filtering of the objects owned by the user when a query is made. For this reason, the cost of filtering takes up a significant portion of the query-processing cost. Furthermore, the existing query-processing scheme processes queries by distributing them throughout the entire network by communicating to all available nodes in the query dissemination section for query processing. Because queries are processed throughout the entire network, this leads to high query processing costs due to unnecessary communication costs. When users designate the query-dissemination range themselves, and if there is an object outside of the query dissemination range, this object will not be included in the query processing even if it is better than the objects that are within the query-dissemination range. This leads to reduced accuracy. Finally, if the same query must continuously be processed at the user's request, the existing query-processing scheme will perform the same query-processing task even if it is within the same range, resulting in increased query-processing costs.

## 3. The Proposed Skyline Query-Processing Scheme

### 3.1. Overall Architecture

In this paper, we propose a new skyline query-processing scheme that can reduce query-processing costs and improve accuracy in the mobile P2P service over DTN. As mentioned, the existing filtering scheme handles all filtering processes after a query is received, resulting in high communication costs. To resolve this problem, the proposed scheme reduces the filtering costs by distributing data and establishing a local skyline before a query is received. Furthermore, in the query-dissemination range expansion policy, specific objects are used from each node's query processing results and the query dissemination range is expanded so that more objects are utilized in query processing, thereby improving its accuracy.

Figure 1 shows the proposed query-processing environment. Each node is a mobile device with mobility, and collects new objects through short-range wireless communication with neighboring nodes. Each node manages neighboring nodes with its communication range and disseminates new objects to its neighboring nodes. When data are disseminated, nodes generate a candidate filtering object by establishing a local skyline considering only the static properties of the objects that are owned by other nodes based on the objects that they themselves own. This leads to a reduction in filtering object generation costs that are incurred when queries are processed. However, the proposed query-processing scheme has a lower query-processing accuracy owing to the limited query-dissemination range. To resolve this issue, in the query-dissemination range expansion policy, after each node processes local skyline queries, the acquired results are reselected through the query-dissemination range and the query-dissemination range is expanded.

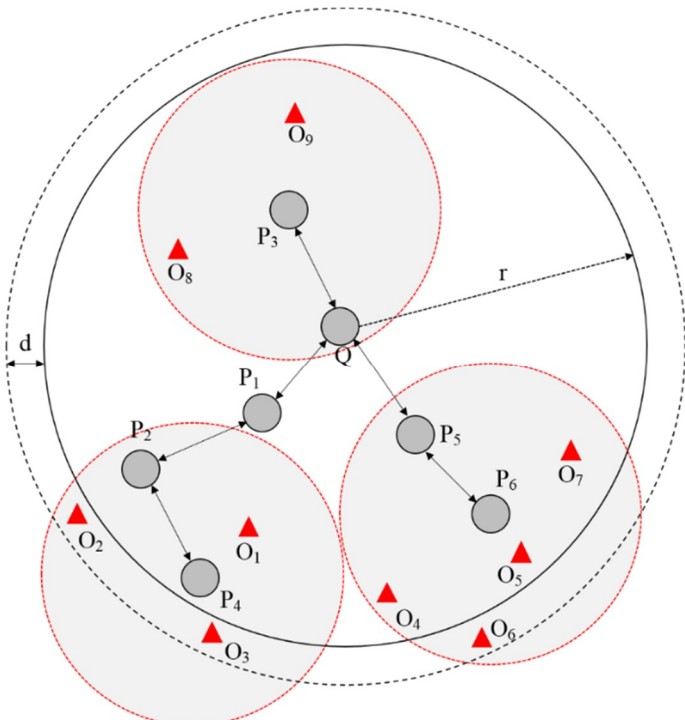

**Figure 1.** Proposed query-processing environment.

Figure 2 shows the overall query processing procedure. In the preprocessing step, data are disseminated each time a node receives a new object before processing a query, and when the data dissemination is complete, a candidate filtering object group is generated by establishing a local skyline. DTN, which cannot guarantee end-to-end connectivity, transmits data through store and forward based multi-hop communication. When performing P2P-based query processing between nodes with mobility in DTN environments, query response time is not guaranteed. Therefore, a query node should set up the query-dissemination range when generating the query. When a query node makes a query with its dissemination range $r$, the query node processes the local skyline query, and the query is disseminated to the node in their communication range. A regular node that receives the query decides on processing the query according to whether or not it is within the query-dissemination range, and the local skyline query is processed accordingly. The regular node disseminates the query to neighboring nodes and waits to receive the results from the child node. Once the node receives the results, the node merges the results and delivers them to the parent node. Finally, the results from all nodes that processed the query are gathered for the query node. The query node merges all results and returns the initial skyline query-processing results to the user by processing a global skyline query. Global skyline query processing refers to the skyline query processing for the objects of all nodes who participated in the query within the network.

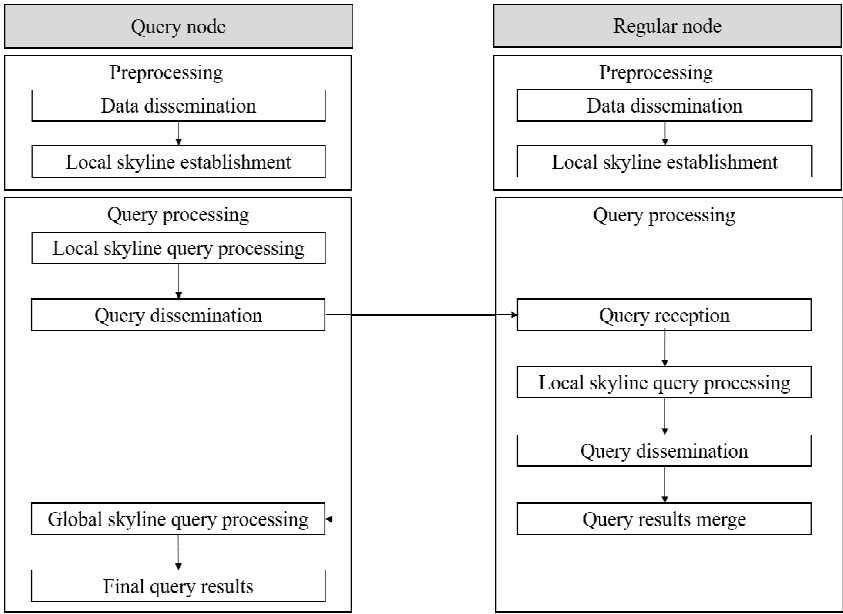

**Figure 2.** Overall query-processing procedure.

## 3.2. Data Structure

All nodes can become query nodes, and the remaining nodes have mobility and can thus receive new object data according to their movements. Each node manages different tables such as object data or query and processing results in its local storage, as shown in Figure 3. Each node manages their queries and query data in the query table (QT), and the query results are managed in the query-processing results table (QRT). Data on nearby nodes are stored in the neighbor node table (NNT) through continuous communication with nearby nodes, and object data received from objects are stored in the owned object table (OT). Nodes establish a local skyline based on object data that they own, and acquired results are merged with candidate-filtering objects and stored in the candidate filtering object table (CFOT). The filtering object groups that are actually used are stored and managed in the filtering object table (FOT).

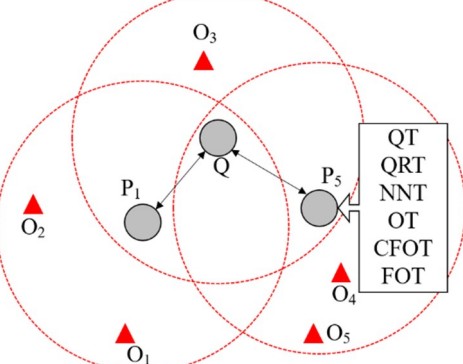

**Figure 3.** Tables managed by each node.

The QT is used to store query data that they themselves generated and query data that were received from other nodes. The query table is composed of <*query_id*, *query_node_id*, *query_node_location*, *query_range*, *query_condition*, *parent_node_id*, *child_node_id*>. The *query_id* is the query identifier, *query_node_id* is the query node identifier, *query_node_location* is the query node location, *query_range* is the query dissemination range, *query_condition* is the query condition, *parent_node_id* is the parent node identifier, and *child_node_id* is the child node identifier.

Local skyline query-processing results are stored in the QRT after each node processes the local skyline. The QRT is composed of <*query_id*, *result_list*>, where *result_list* stores the query results in list form and stores them as values of each condition that corresponds to *query_condition*. The NNT

stores data on neighboring nodes. The proposed scheme enables the node to communicate with other nodes within their communication range and stores the data on communicating nodes in the NNT in order to assess who the neighboring nodes are. The NNT is composed of *<node_id, node_location, node_vector, query_id_list>*. The *node_id* is the neighboring node identifier, *node_location* is the neighboring node location, *node_vector* is the neighboring node's movement speed, and *query_id_list* shows the *query_id* data that are currently saved.

The OT stores the object data received from objects near the node. The object table is composed of *<object_id, object_location, object_condition, object_update_time>*, where *object_id* is the object identifier, *object_location* is the object location, *object_condition* is the object's property value, and *object_update_time* is the time when the object data were received. The CFOT stores objects that were acquired by establishing a local skyline. The CFOT is composed of *<object_id, object_location, object_condition, object_update_time>*, where *object_id* is the object identifier, *object_location* is the object location, *object_condition* is the object's property value, and *object_update_time* is the time when the object data were received. The FOT stores objects merged with existing initial filtering objects and local skyline query-processing results after processing the local skyline queries. The FOT is composed of *<object_id, object_location, object_condition, object_update_time>*, where *object_id* is the object identifier, *object_location* is the object location, *object_condition* is the object's property value, and *object_update_time* is the time when the object data were received.

### 3.3. Local Skyline Processing

Each node receives new objects data based on their movements, and data dissemination begins when the new object data are received. The proposed scheme can maintain accuracy during data dissemination because the object data from a disconnected node are indirectly owned by the other nodes. Therefore, its object data can be reflected during query processing through nodes who once received the data from nearby even if the node gets disconnected while a query is being processed.

Figure 4 shows how $P_1$ and $P_2$ that are nearby each other disseminate the object data that they own. In the proposed scheme, when nodes first communicate with a specific node, all the object data that they own are disseminated. After the initial data dissemination to a specific node, new object data are disseminated to that node only when the previous data have been received. When $P_1$ and $P_2$ communicate for the first time, they transmit the object data that each of them owns. However, when each node transmits object data to another node, this may lead to an issue where duplicate data are stored because the object data already owned were transmitted. To solve this problem, the node that is transmitting the object data will compare the data acquired through the data dissemination with the object data that the node already owns, and only non-duplicate object data are stored. When the data dissemination is complete, because the node that was distributing data has now received new object data, these new objects are disseminated as the node re-disseminates data to all nearby nodes.

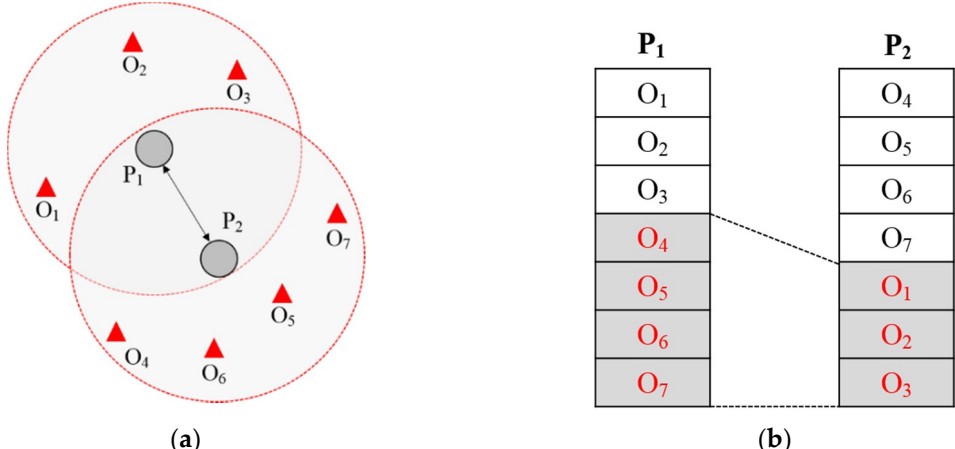

(a)                                                                 (b)

**Figure 4.** Data dissemination between $P_1$ and $P_2$: (**a**) Communication between $P_1$ and $P_2$; (**b**) Data dissemination between $P_1$ and $P_2$.

The data dissemination step involves distributing objects that the node owns to other nearby nodes. Each time a node receives objects from a nearby node, the newly received object data are sent through *MSG_OBJECT_DISSEMINATION* in order to disseminate the received object to nearby nodes. The node receiving the *MSG_OBJECT_DISSEMINATION* will determine whether the node already owns the disseminated data. If there are object data that the node does not own from among the disseminated objects, they are stored in the *OBJECT_TABLE*. Algorithm 1 shows the node processing algorithm when the *MSG_OBJECT_DISSEMINATION* is received during the data-dissemination process. When nodes receive the *MSG_OBJECT_DISSEMINATION* message as shown in the algorithm, they will first assess whether or not they are within the query-dissemination range. If they are not within the query-dissemination range, they will send *MSG_QUERY_BROADCAST_FAIL* to the node who sent the query to them. They will then compare the *object_id* of all objects in their *OBJECT_TABLE* with the *object_id* of *MSG_OBJECT_DISSEMINATION*, and determine if the same *object_id* exists. If there is an object that does not exist, it will be stored in their *OBJECT_TABLE*.

---

**Algorithm 1** receive_MSG_OBJECT_DISSEMINATION()

---

```
{
    if (Node_Location<Query_Range)
        for (OBJECT_TABLE.size()){
            if (MSG_OBJECT_DISSEMINATION.object_id   does   not   exist   in
OBJECT_TABLE)
                store the received object information in OT}
}
```

---

In the proposed scheme, query-processing costs are reduced by establishing a local skyline before queries are processed. This will reduce filtering costs that are incurred during query processing. When all nodes complete the data distribution, they generate a candidate filtering object group by establishing a local skyline based on the object data that they own. Figure 5 shows how candidate object groups are generated by building a local skyline. $P_1$ generates a candidate object group through a skyline based on the static properties of the object data it owns, except for the distance property as shown in Figure 5a. Although the distance property of each object is a property that is mapped through the distance from the query node corresponding to the user device transmitting the queries, that property is excluded because it cannot be determined, as local skylines are established before queries are even received. The other remaining properties are static properties that do not change. A local skyline is established by considering the following two static properties to process the skyline. Establishing a local skyline is a scheme that sorts only objects that are not dominated by other objects in the same way as in regular skyline processing. $O_1$, $O_2$, and $O_4$, which are objects that are not dominated as shown in Figure 5b, comprise a candidate-filtering object group by establishing a local skyline.

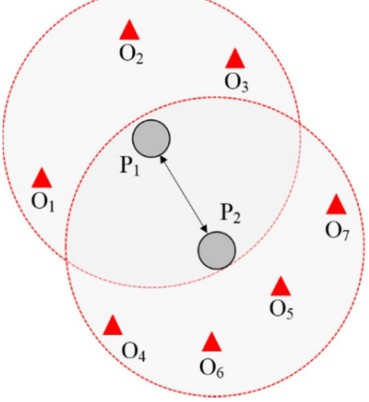

| Object | Price | Quality | Distance |
|--------|-------|---------|----------|
| $O_1$ | 3,500 | 4 | |
| $O_2$ | 2,100 | 2 | |
| $O_3$ | 5,000 | 5 | |
| $O_4$ | 2,000 | 2 | |
| $O_5$ | 3,600 | 4 | |
| $O_6$ | 4,000 | 2 | |
| $O_7$ | 4,800 | 3 | |

(**a**)

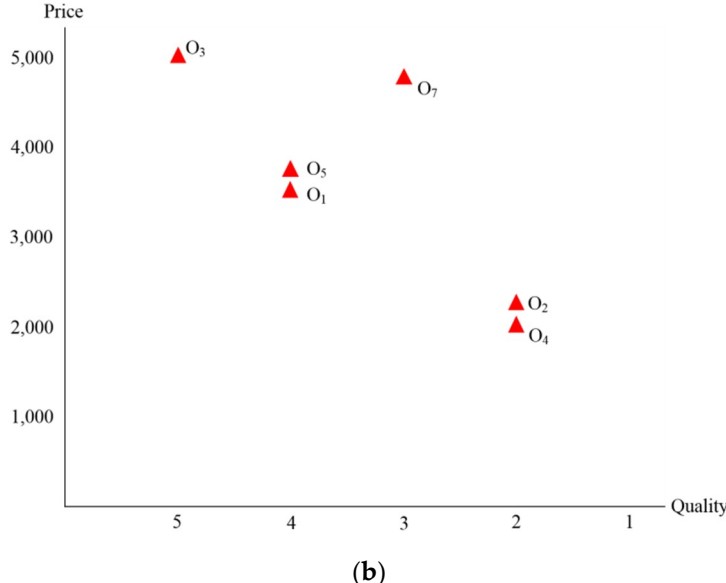

**Figure 5.** Generating a candidate filtering object group by establishing a local skyline: (**a**) object information stored in $P_1$; (**b**) local skyline.

In the existing filtering scheme, filtering objects are generated only after a query is received in order for filtering to occur. For this reason, filtering costs are added to processing costs and so query processing performance is reduced. By contrast, the proposed scheme reduces the cost of generating filtering objects because it generates candidate-filtering objects by establishing a local skyline before queries are received. Algorithm 2 shows the algorithm for establishing a local skyline. Nodes can establish a local skyline based on the object data in their own *OBJECT_TABLE* after the data distribution. Only the object's static properties are considered in establishing a local skyline. Objects that are not dominated by the node's other objects are acquired from the results of establishing the local skyline, and they are stored in *Candidate_Filtering_Object_Table*.

---

**Algorithm 2** process_Pre_Skyline()

---

{

    **for** (OBJECT_TABLE.size()){

        **if** (all static properties of object i > all static properties of

object j)

            store i in candidate filtering object set;

    }

}

---

### 3.4. Global Skyline Processing

Global skyline processing generates the final query results by merging the local skyline generated by each node in the query-dissemination range. The query node disseminates the issuing query to the nodes within communication range to collect and merge the local skyline generated each node. When a query is received from the user, each node performs filtering by using specific objects from the candidate filtering object group that the node owns. The purpose of filtering is to reduce the number of messages that are generated when queries are processed because the number of calculations increases with the number of objects.

Table 1 provides the distance properties that were mapped by calculating the distance between query nodes based on the object data that are owned by the node. In the same way, the distance properties of each object are calculated and mapped in the candidate-filtering object group as given in Table 2. Here, the distance property refers to the distance between the query node and the object it owns. The query node selects the farthest object out of the candidate-filtering object group as the

first filtering object. $O_4$, which has the farthest distance, is selected as the first filtering object. The node selects the nearest $O_4$ from the candidate filtering object groups as the first filtering object. This is because $O_5$ was dominated by $O_1$ during the process of establishing a local skyline, but it was no longer dominated by $O_1$ after the query was received and the distance property was determined; hence, it became an object that could be selected as part of the global skyline query-processing results. Table 3 shows the properties of $P_1$'s first filtering object. When the first filtering object is selected as such, the nodes begin filtering by including the distance properties based on the objects they own. $O_1$, which was selected as the first filtering object, filters $O_6$ and $O_7$, which are objects with a lower value for all properties.

**Table 1.** Objects owned by $P_1$.

| Object | Price | Quality | Distance |
|--------|-------|---------|----------|
| $O_1$ | 3500 | 3 | 4 |
| $O_2$ | 2100 | 2 | 2 |
| $O_3$ | 5000 | 5 | 3 |
| $O_4$ | 2000 | 2 | 6 |
| $O_5$ | 3600 | 4 | 4 |
| $O_6$ | 4000 | 2 | 5 |
| $O_7$ | 4800 | 3 | 8 |

**Table 2.** Candidate filtering object group of $P_1$.

| OID | Price | Quality | Distance |
|-----|-------|---------|----------|
| $O_1$ | 3500 | 3 | 4 |
| $O_3$ | 5000 | 5 | 3 |
| $O_4$ | 2000 | 2 | 2 |

**Table 3.** First filtering object of $P_1$.

| Object | Price | Quality | Distance |
|--------|-------|---------|----------|
| $O_1$ | 3500 | 3 | 4 |

In the filtering and local skyline query processing step, after the node receives a query, the first filtering object is selected from the *Candidate_Filtering_Object_Table* that was obtained from the local skyline. When the node receives a query, the distance property is first matched by calculating the distance between the query node and the object in order to determine the distance property of the object in *Object_Table*. Equation (1) is used to calculate the distance between the query node and the object, and the distance property is determined by calculating the distance using this formula:

$$value_{dist} = \sqrt{(Q_x - O_x)^2 + (Q_y - O_y)^2}, \tag{1}$$

where, $Q_x$ and $Q_y$ respectively show the x and y coordinates corresponding to the query node's location, while $O_x$ and $O_y$ show the x and y coordinates respectively, corresponding to the object's location. This is the same as the general distance calculation formula between two coordinates. Because a normal node receives the query node's location data along with the query itself, all nodes that receive the query can use Equation (1) to calculate the distance.

Algorithm 3 shows the algorithm for selecting the first filtering object. Each node determines the distance property of the objects that the node owns using Equation (1). The node selects the object with the largest distance property from the candidate-filtering object group as the first filtering object. The first filtering object is stored in the *Filtering_Object_Table*.

---

**Algorithm 3** select_First_Filtering_Object

{
　　**for** (Candidate_Filtering_Object_Table.size()){
　　　map $value_{dist}$ to the distance property of each object;

---

> **if** (the object i has the largest distance among the filtering objects
>
>            select the object i as the first filtering object;
>
>     }
>
> }

Algorithm 4 shows the algorithm for filtering through the first filtering object. When the first filtering object is selected, the nodes will filter the object that they have, which will take part in the query processing. The filtering target will be an object with property values including the distance property, which is a dynamic property, that are all lower than those of the first filtering object.

---
**Algorithm 4** process_Filtering
---

{

    **for** (OBJECT_TABLE.size()){

        **if** (all properties of each object in OBJECT_TABLE are lower than Filtering_Object)

            filter object;

    }

}

---

Algorithm 5 shows how the local skyline queries are processed. After filtering, the node will process the local skyline queries for the remaining objects that were not filtered. The results obtained from processing the local skyline queries are stored in the *Query_Result_Table*. Furthermore, the filtering object group is adjusted by merging the local skyline query-processing result with the filtering result group. The node removes the remaining objects after the filtering, except for those that are not dominated by the objects that the node owns. Objects that are not removed are stored in the *Query_Result_Table*.

---
**Algorithm 5** process_Local_Skyline
---

{

    **for** (OBJECT_TABLE.size()){

        **if** (all properties of object i> all properties of object j)

            remove j from OBJECT_TABLE;

    }

    store unremoved objects to Query_Result_Table;

}

---

The proposed scheme also proposes a policy for expanding the query-dissemination range. In this policy, more objects can be included in the query processing by recalculating the query dissemination range to disseminate queries through an expanded query-dissemination range. This policy is for improving the accuracy. In the existing distribution range, even nodes within the range are unable to process queries because they are unable to communicate. However, expansion can allow such nodes to go beyond the existing query-dissemination range so that they will be able to communicate. For the query processing, the user specifies the query-dissemination range, and queries are processed within a specific range. Therefore, although more objects are required to process queries from the perspective of accuracy, because nodes who are outside the query-dissemination range are unable to participate in processing queries, the accuracy of query processing will suffer. Each node processes local skyline queries for the object data that the node owns, and then generates a new filtering object group based on the results. Local skyline query processing means that skyline queries are processed based on the object data that are owned by the node. If the object with the farthest distance from the new filtering object group is outside the existing query dissemination range, the range will be reselected based on the distance property value of that object.

Figure 6 shows how the query dissemination range is expanded. $P_5$ merges $O_1$, $O_7$, and $O_8$, which are part of the local skyline query-processing results, with the existing filtering objects after processing the local skyline queries. Because $O_8$, which has the farthest distance among the newly generated filtering object group, is farther outside of the existing query-dissemination range, this

range will be increased by $d$ based on $O_8$. In the existing query-dissemination range, even if $P_5$ disseminates a query to $P_7$, $P_7$ does not need to process the query because it is not within the query dissemination range. Thus, if $P_7$ has an object that can influence the skyline query-processing results, the accuracy will be decreased. However, if the query-dissemination range is expanded through $O_7$, which is an object owned by $P_5$, $P_7$ will thus be included in the query, and the issue of reduced accuracy will be resolved. Table 4 shows the filtering object group after the local skyline query of $P_5$ is processed. $P_7$ is able to participate in the query when the query-dissemination range expands from $P_5$, and it can perform filtering and local skyline query-processing accordingly.

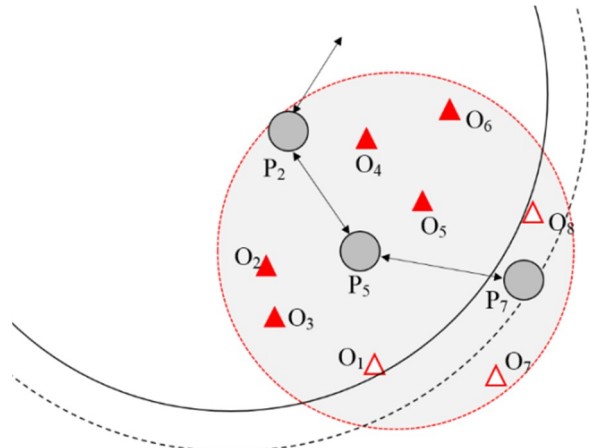

**Figure 6.** Query-dissemination range expansion through $P_5$.

**Table 4.** Local skyline query processing result of $P_5$.

| Object | Price | Quality | Distance |
|--------|-------|---------|----------|
| $O_{11}$ | 3,500 | 4 | 8 |
| $O_{13}$ | 5,000 | 5 | 7 |
| $O_{14}$ | 2,000 | 3 | 8 |
| $O_{17}$ | 4,800 | 5 | 15 |

The query-dissemination range expansion and query dissemination is the step where queries are transmitted to neighboring nodes after the *Query_Result_Table* is generated. After processing the local skyline query, the node assesses whether there is an object from the local skyline query-processing results with a distance property that is larger than the existing query-dissemination range. If such object exists, the query-dissemination range is reselected to match the distance property of that object. Algorithm 6 shows the algorithm for expanding the query-dissemination range based on the local skyline query-processing results. Each node reselects the query dissemination range based on the distance property of an object in the *Query_Result_Table*, which is greater than the *Query_Range*.

---

**Algorithm 6** process_Expend_Query_Range

---

```
{
    for (Query_Result_Table.size()){
        if    (distance    of    the    Query_Result
    >Query_Range)
            re-select the range of query dissemination;
    }
}
```

---

After the query dissemination range is expanded, the node will disseminate *MSG_QUERY_BROADCAST* to neighboring nodes in the *Neighbor_Node_Table*.

*MSG_QUERY_BROADCAST* is composed of *<node_id, query_id, query_node_id, Query_Result_Table, query_range>*.

In the proposed scheme, not all the local skyline results of each node are returned to query nodes by merging the results. However, the results are merged by returning them to the parent node instead. Through this process, only the optimized results are returned to the query node. Hence, the query node can merge only the optimized results to process the global skyline queries, thereby preventing device load. Furthermore, this reduces the number of messages that are generated as the results are returned, and therefore, the query-processing performance is improved.

Figure 7 shows how the results are merged and returned. The node's local skyline query processing results from $P_1$ to $Q$ are returned to the parent node. The parent node then merges the results obtained from the child node with its own local skyline query-processing results, and the new results are returned to its own parent node. The results are merged in the same way as in the local skyline query processing. Here, the child node refers to the node who received the query that was disseminated, and parent node refers to the node who transmitted the query. When the global skyline queries are processed, the local skyline query-processing results from all nodes involved in the query are returned to the query node. The query node then generates the initial skyline query-processing results by processing the global skyline queries based on the local skyline query-processing results from all nodes involved in the query. Here, the global skyline query is processed in the same way as processing a local skyline query, but the object that is the target of query processing is one that is part of the local skyline query-processing results of the nodes that took part in the query.

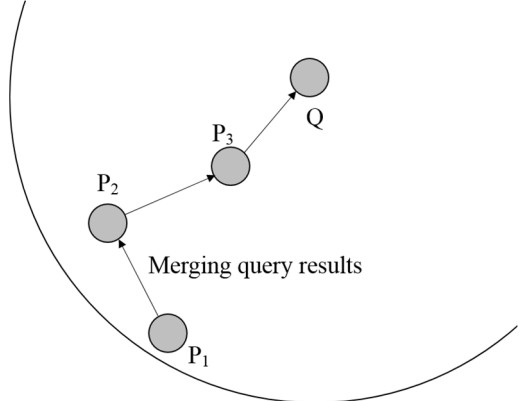

**Figure 7.** Merging results from $P_1$ to $Q$.

Algorithm 7 shows the algorithm for merging and returning the results. The node who received the results from the child node stores all the results into the node's *Query_Result_Table*. The local skyline queries are processed for the stored results, and new result groups are generated. The local skyline query-processing results that were obtained based on the returned results and its own results replace the results in the *Query_Result_Table*. The node delivers these results to the parent node through a *MSG_RESULT_BROADCAST* message.

---

**Algorithm 7** process_Result_Merge_and_Return

---
{
    **if** (MSG_RESULT_BROADCAST is received from the child node)
      store the returned object in Query_Result_Table;
      **for** (Query_Result_Table.size()){
        **if** (all properties of object i> all properties of new object j)
          remove j from OBJECT_TABLE;
      }
      replace   unremoved   objects   with   results   from
Queries_result_Table;
      send MSG_RESULT_BROADCAST to a parent node;

---

```
    }
```

When the query node receives all the local skyline query-processing results through the *MSG_RESULT_BROADCAST* message, the final global skyline query is processed and the first query-processing results are generated in this step. When the query node receives all objects from the child nodes, the global skylines are performed just as how normal nodes would re-perform a local skyline by merging the results. The final results obtained through the global skyline are sent to the user.

## 4. Performance Evaluation

We demonstrate the superiority of the proposed scheme through performance comparison with VDR [30] and DPL [31]. The query-processing performance was evaluated by changing the number of nodes, the number of objects, communication range, query-dissemination range, and movement speed. JAVA was used in a computer running on Intel Core i3 CPU with 3.07 GHz and 8G RAM for the performance evaluation. Table 5 shows the parameters used for the performance evaluation.

**Table 5.** Performance evaluation environments.

| Parameter | Value |
|---|---|
| # of nodes | 500~3500 |
| # of objects | 100~700 |
| Communication range (m) | 15~45 |
| Range of query dissemination (m) | 200 × 200 |
| Moving velocity (m/sec) | 10 |

When queries are processed in DTN, each node will communicate with many nodes if there are more nodes in the query range, and the amount of object data received by each node will increase if there are more objects. Therefore, queries are processed for a large number of objects. Moreover, because nodes receive object data within a wide range according to the expansion of the communication range, even more object data can be received. Hence, the number of messages that are generated during communication will increase.

To compare the performance between the proposed and existing schemes, the number of messages generated based on changes in the number of nodes, number of objects, and communication range was examined. To evaluate the number of messages generated according to changes in the number of nodes, the number of nodes was changed from 500 to 3500 while the number of objects was set to 100, communication range to 15 m, query dissemination range to 200 m × 200 m, and movement speed to 10 m/s. As shown in Figure 8, more objects could be filtered in advance even in a network environment with a distribution of many nodes by establishing a local skyline to reduce communication costs, thereby reducing the number of messages by 18% to 84% compared to the existing schemes.

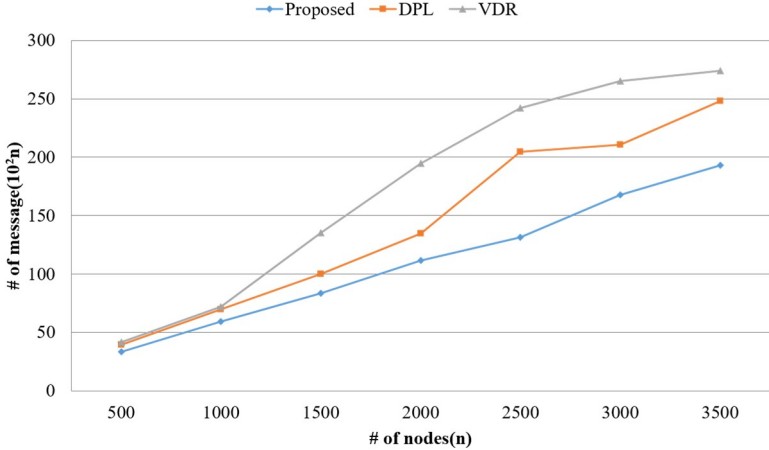

**Figure 8.** Number of messages generated according to the number of nodes.

To evaluate the number of messages generated according to changes in the number of objects, the number of objects was changed from 100 to 700 while the number of nodes was set to 500, communication range to 15 m, query-dissemination range to 200 m × 200 m, and movement speed to 10 m/s. As shown in Figure 9, even if the number of objects increases, more objects can be filtered in advance by establishing a local skyline. The increase in the number of generated messages is smaller and the number of messages generated decreases by 18% to 454% compared to the existing schemes.

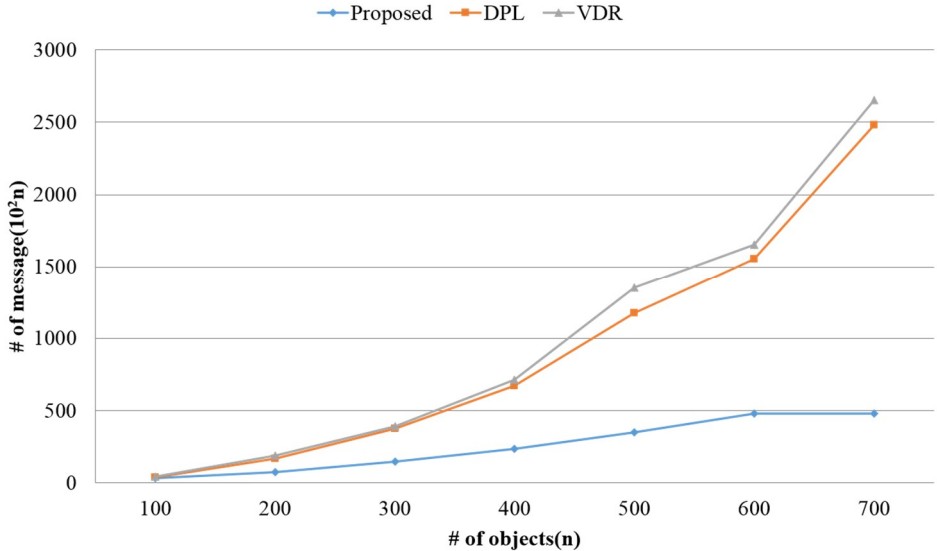

**Figure 9.** Number of messages generated according to the number of objects.

To evaluate the number of messages generated according to changes in the communication range, the communication range was changed from 15 m to 45 m while the number of nodes was set to 500, objects to 100, query-dissemination range to 200 m × 200 m, and movement speed to 10 m/s. As shown in Figure 10, communication with more objects is possible based on the size of the node's communication range in the proposed scheme. Hence, more objects can be filtered by establishing a local skyline, and the performance is improved by 18% to 299% compared to the existing schemes.

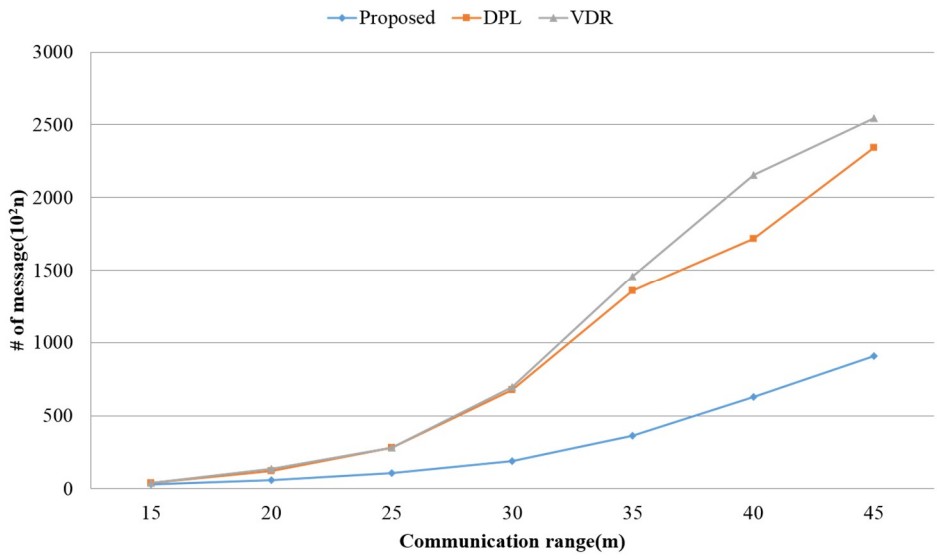

**Figure 10.** Number of messages generated according to communication range.

When queries are processed in DTN, if there are many nodes in the query range, the number of nodes that each node can communicate will increase. The number of nodes that can participate in

processing the query will also increase, and the query-processing accuracy will improve. Because each node will have to process queries for more objects as the number of objects increases, the query-processing accuracy will improve accordingly. Furthermore, when each node's communication range increases, the query-processing accuracy will improve because more data will be received from more objects. The query-processing accuracy is calculated Equation (2):

$$\text{Accuracy} = \frac{RR}{TR}, \tag{2}$$

where $TR$ is the query result set to be generated and $RR$ is the query result set returned through the query processing.

The query-processing accuracy was evaluated according to the number of nodes, number of objects, and communication range in order to compare the performance of the proposed scheme to the existing schemes. To evaluate the query-processing accuracy according to changes in the number of nodes, the number of nodes was changed from 500 to 3500 while the number of objects was set to 100, communication range to 15 m, query-dissemination range to 200 m × 200 m, and movement speed to 10 m/s. As shown in Figure 11, because more nodes participated in the query through the query-dissemination range expansion policy, the query-processing objects increase, resulting in 2% average increase in accuracy compared to the existing schemes.

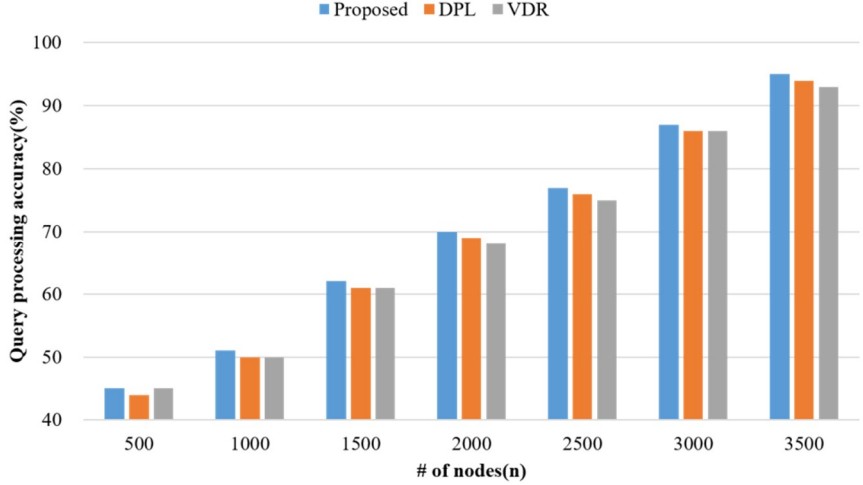

**Figure 11.** Query-processing accuracy according to the number of nodes.

To evaluate the query-processing accuracy according to changes in the number of objects, the number of objects was changed from 100 to 700 while the number of nodes was set to 500, communication range to 15 m, query-dissemination range to 200 m × 200 m, and movement speed to 10 m/s. As shown in Figure 12, because more nodes receive more objects through the query-dissemination range expansion policy, there is 4% average increase in accuracy compared to the existing schemes.

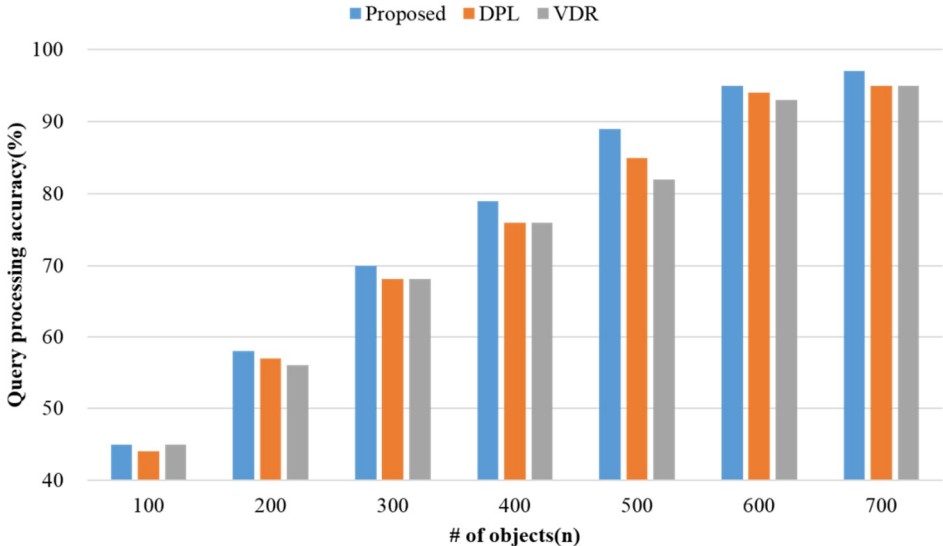

**Figure 12.** Query-processing accuracy according to the number of objects.

To evaluate the query-processing accuracy according to changes in the communication range, the communication range was changed from 15 m to 45 m while the number of nodes was set to 500, objects to 100, query-dissemination range to 200 m × 200 m, and movement speed to 10 m/s. As shown in Figure 13, more object data that are disseminated in the network could be evaluated based on the size of the node's communication range, and there is 1% average increase in accuracy compared to the existing schemes.

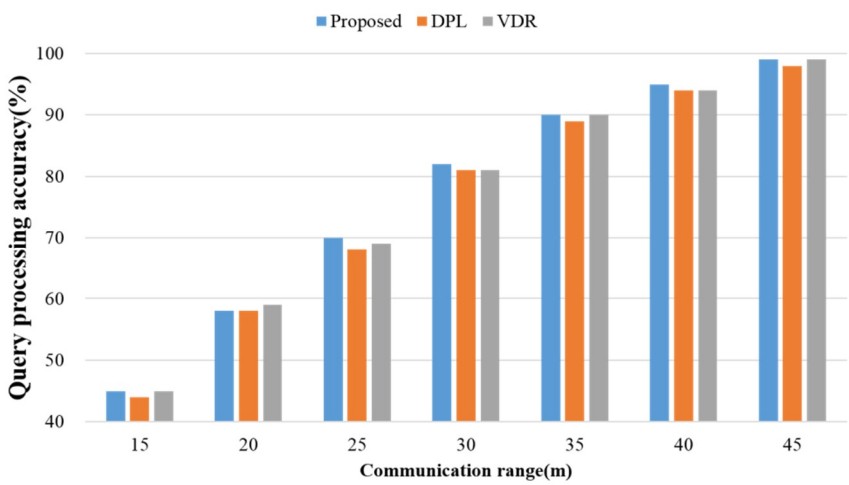

**Figure 13.** Query-processing accuracy according to communication range.

As the node's communication range increases, nodes can receive data from more objects, and they can communicate with more nodes. As a result, the query-processing accuracy improves. However, the number of messages generated increases, causing an increase in query-processing costs. The query-processing time was evaluated according to the number of nodes, number of objects, and communication range in order to compare the performance of the proposed scheme to the existing schemes. To evaluate the query-processing time according to changes in the number of nodes, the number of nodes was changed from 500 to 3500 while the number of objects was set to 100, communication range to 15 m, query-dissemination range to 200 m × 200 m, and movement speed to 10 m/s. As shown in Figure 14, as the filtering process cost in the actual query-processing

decreases by establishing the local skyline, the processing time also decreases by 4% to 146% compared to the existing schemes.

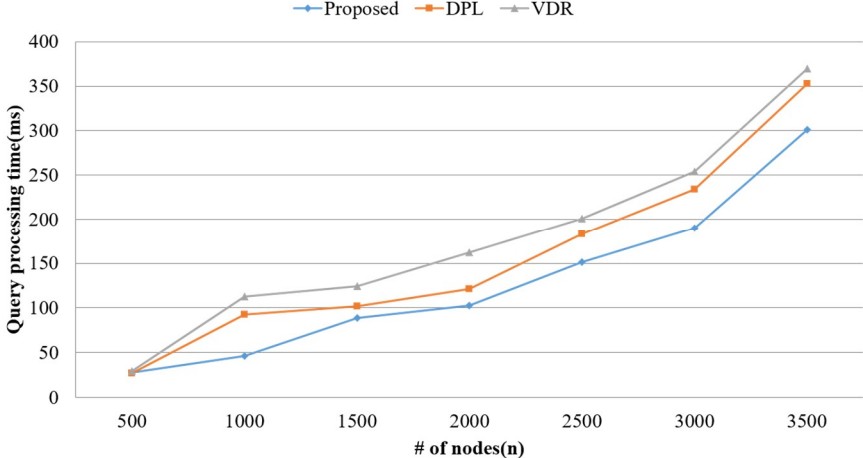

**Figure 14.** Query-processing time according to number of nodes.

As shown in Figure 15, as more objects are filtered by establishing a local skyline, and because queries are processed based on objects that are owned after filtering, the query-processing time decreases by 4% to 642% compared to the existing schemes.

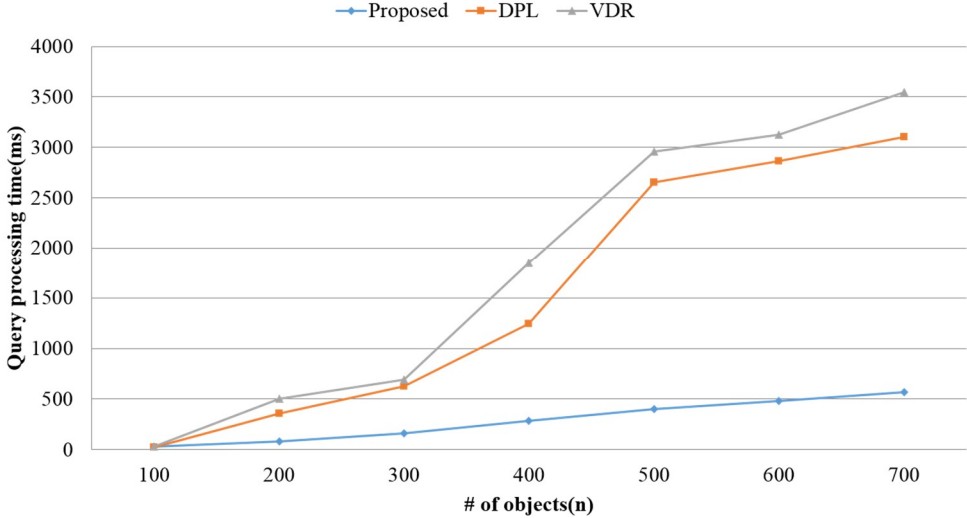

**Figure 15.** Query-processing time according to number of objects.

As shown in Figure 16, more objects are filtered by establishing a local skyline based on the size of the node's communication range. Thus, the query-processing time is reduced by 4% to 48% compared to the existing schemes.

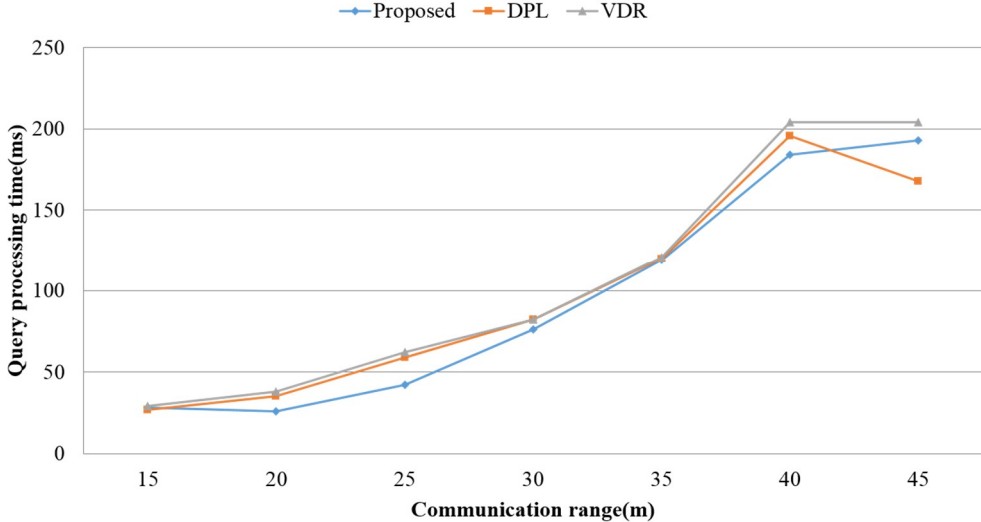

**Figure 16.** Query-processing time according to communication range.

## 5. Conclusion

Various query-processing schemes are being studied in line with advancements in P2P location-based services that use diverse wireless communication technologies. Among them, skylines that process queries by considering different properties have been a topic of interest in these studies. Many filtering schemes have been used in previous studies in an effort to improve the query-processing performance in DTN. In this paper, we proposed a P2P-based efficient skyline query-processing scheme to improve the query-processing performance and accuracy over DTN. The proposed scheme proposes a local skyline processing and query-dissemination range expansion policy. In the existing VDR and schemes, filtering occurs at the point where queries are processed. This leads to several issues where filtering costs are added to query-processing costs, and query-processing performance suffers. In this paper, to reduce the filtering cost during query processing, candidate filtering objects are generated by establishing a local skyline based on the static properties of objects before a query is received, thereby reducing the filtering costs. Furthermore, the query-processing accuracy is improved owing to the expansion of the query-dissemination range. Based on the performance evaluation conducted, the proposed scheme demonstrated superior results compared to the existing VDR and DPL.

**Author Contributions:** conceptualization, K.B., S.P., J.L, and J.Y.; methodology K.B., S.P., J.L, and J.Y.; software, S.P.; validation, K.B., S.P., and J.L.; formal analysis, K.B. S.P., and J.L; data curation, S.P.; writing—original draft preparation, K.B. and S.P.; writing—review and editing, J.Y.

**Funding:** This work was supported by the National Research Foundation of Korea(NRF) grant funded by the Korea government(MSIT) (No. 2019R1H1A2079843) and by Next-Generation Information Computing Development Program through the National Research Foundation of Korea(NRF) funded by the Ministry of Science, ICT (No. NRF-2017M3C4A7069432)

**Conflicts of Interest:** The authors declare no conflict of interest.

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
