# Peer review of "Mobile P2P-Based Skyline Query Processing over Delay-Tolerant Networks"

_electronics, doi:10.3390/electronics8111276_

Round 1

Reviewer 1 Report

This paper studied the skyline query processing scheme for Delay Tolerant Networks.

The main contribution is a new skyline query processing scheme to improve the query processing performance and accuracy in a mobile P2P service over delay tolerant networks.

The proposed scheme collects data on the query object from neighboring nodes and establishes a local skyline through static properties to reduce query processing costs.

The query dissemination range is expanded by enforcing a query dissemination range expansion, which is able to improve the query accuracy in a non-uniform distribution environment.

Finally, the performance of the proposed scheme beats existing schemes via experiments.

Overall, this paper is stated in a clear way and the contributions are significant. One minor issue: the concept "Skyline Query Processing" should be explained in detail.

I suggest accepting the paper after a minor revision.

Author Response

Dear reviewer,

Thank you for your attentive indications and good comments.

Our paper was partially rewritten in order to reflect and complement your comments.

Please refer to the attached file about the detailed revisions.

Thanks.

Jaesoo Yoo

Reviewer 2 Report

Dear authors, in my opinion, this article brings significant argument regarding the treated subject. The studies and the experimental results are valuable, correctly defined and properly described. I think that your article will be interesting to readers of Electronics journal and useful for some of them.

Minor problems that I see are related to the fact that the article does not follow the provided template:

- All the schemes (figures) must follow the same formatting:

Figures must be placed in an invisible column table - are the figures inserted in this way? If not please correct. Two figures must be placed in an invisible 2-columns table - instead of figure 6 definition! The space before and after the figures must be corrected, set according to the template. Caption of the figures (figures title) should be centered.  The numbering (order number) of the figures it should be bold - see Figure 7.

- It should take the space between the value and the appropriate unit of measure      

 (Ex: 15 m - OK and 15m - NOK) - please correct for all the values.

- The space before and after equations must be corrected, set according to the template. The text following an equation need not be a new paragraph. Please punctuate equations as regular text.

- The numbering (order number) of the tables it should be bold - see table 4.

Best regards,

Author Response

(The authors gave the same response as above.)
